# Colorimetric Visualization of Chirality: From Molecular Sensors to Hierarchical Extension

**DOI:** 10.3390/molecules30081748

**Published:** 2025-04-14

**Authors:** Yuji Kubo

**Affiliations:** Department of Applied Chemistry, Graduate School of Urban Environmental Sciences, Tokyo Metropolitan University, 1-1 Minami-Osawa, Hachioji, Tokyo 192-0397, Japan; yujik@tmu.ac.jp

**Keywords:** colorimetric chiral sensing, molecular organization, chemometrics, helical polymer, nanoprobe

## Abstract

The optical sensing of chirality is widely used in many fields, such as pharmaceuticals, agriculture, food, and environmental materials. In this context, the color-based cascade amplification of chirality, coupled with chiral recognition for analytes, provides a low-cost and straightforward detection method that avoids the use of expensive and sophisticated instrumentation. However, the realization of chiral detection using this approach is still challenging because the construction of a three-dimensional optical recognition site is required to easily discern differences in chirality. Therefore, considerable efforts have been dedicated to developing a hierarchical approach based on molecular organization to provide colorimetric sensors for chirality detection. This review covers function-integrated molecular sensors with colorimetric responsive sites based on absorption, fluorescence, and aggregation-induced emission enabled by molecular organization. In line with the hierarchical approach, data-driven chemometrics is a useful method for quantitative and accurate chiral pattern recognition. Finally, colorimetric nanomaterials are discussed, focusing on sensing platforms using noble-metal nanoparticles, carbon dots, and photonic crystal gels.

## 1. Introduction

Simplicity, cost-effectiveness, and fast response are the key properties for sensors. Colorimetric sensors meet these criteria; in particular, a sensing method based on visual color changes is suitable for use in a wide range of applications, such as on-site environmental monitoring, point-of-care tests in a clinical environment, and food safety testing [1]. For example, pH test strips with a dye indicator are widely employed to detect the pH of solutions based on their colorimetric response to the tested solution. However, the low anti-interference ability of colorimetric sensors limits their use in highly sensitive and quantitative analysis. Therefore, sophisticated designs and concepts are particularly desirable for colorimetric sensors for chiral identification, which is inevitably required for many applications in pharmaceutical, food, cosmetic, and agricultural industries [2]. Notably, because approximately half of the small molecules used in medicine are chiral, the manufacturing of unichiral medications is important for the pharmaceutical industry [3]. The crucial role of chirality has led to increasing demand for pure enantiomeric drugs and related compounds, making it imperative to identify the chirality of molecules. Among the optical sensing techniques, fluorescence probes are the simplest and most rapid detection methods for chiral analytes, and they are widely employed in various research fields [4]. However, fluorescence sensors with a single signal output are often susceptible to interference *via* the environment and, in some cases, exhibit poor reproducibility; therefore, ratiometric fluorescent probes, in which two fluorescence signals are simultaneously measured at different wavelengths and the intensity ratio is calculated, have been developed to reduce background interference [5,6]. Because numerous insights into organic colorants related to dye chemistry have been exploited to develop dye-based functional materials [7], the visual identification of chirality through probe-based colorimetric response is another promising approach. It is important to note that the operationally simple and rapid identification of the stereogenic properties of analytes that do not require expensive instruments and skilled users faces synthetic difficulties in integrating the functions required for the colorimetric sensing of chirality in a single discrete molecule, motivating the exploitation of hierarchical organization according to a bottom-up approach.

This review article discusses the colorimetric detection of chiral molecules by optically active host systems. The author first describes single-host-based colorimetric chiral recognition. The synthetic limitations of this approach motivated researchers not only to prepare molecular ensembles but also to design aggregation-induced emission (AIE)-based chiral sensing to avoid tedious preparation. Additionally, data-driven chemometrics such as multivariate analysis, being a statistical method that examines relationships among three or more variables, has been introduced to improve the quantitative analysis of such sensing. An alternative organization approach involving liquid crystals and polymer scaffolds is also discussed. Finally, colorimetric detection using nanomaterials, such as noble-metal nanoparticles, carbon dots, and molecularly imprinted photonic crystals, is reviewed in the context of hierarchical colorimetric sensing systems.

## 2. Design Principle

Chiral sensors are composed of “recognition” and “optical signaling” units, which must communicate with each other to attain effective translation and amplification of chirality (Figure 1a,b). Therefore, careful design and strategies are highly desired; distinguishing non-superimposable mirror images of the target analytes requires the construction of a three-dimensional recognition site for the analytes that should communicate with the optical signaling sites. However, the synergistic integration of these functions in molecular systems is synthetically difficult. An accurate determination of the purity and enantiomer excess (*ee*) of chiral analytes with performance comparable to that of chromatographic techniques still requires significant research. Inspired by biomolecular systems, hierarchical bottom-up strategies are powerful approaches to the fabrication of binary co-assemblies and template-mediated assembly (Figure 1c–e). This approach involves the controlled assembly of constituent molecular modules, which can lead to enhanced synergistic functional outcomes [8]. Indeed, molecular chirality can be transferred to supramolecular systems through well-tailored self-assembly processes [9]. This insight is beneficial for the exploration of colorimetric sensors embedded into one-, two-, and three-dimensional platforms.

## 3. Small Molecule-Based Chromogenic Sensors

In early work, environment-sensitive chromophores were attached to chiral recognition units. Azophenol dyes are some of the traditional chromophores, and they are responsive to chemical stimuli such as acids and bases. Kaneda et al. synthesized chiral azophenol crowns [10]. When chiral amines were tested as putative guests, the difference in the *λ*_max_ values of at most 11 nm was observed between the diastereomeric sets of the guests, which makes it difficult to determine chirality. In a breakthrough study, Kubo et al. incorporated indophenol dyes into the 1,1′-binaphthyl unit on the calix[4]crown platform to produce (*S*)-**CCI** (Figure 2a) [11]. 1,1′-Binaphthyl units with *C_2_* axial chirality and high inversion barriers of 23.8 kcal mol^−1^ [12], are versatile building blocks not only for asymmetric-synthesis catalysts but also for the fluorophore units of chemosensors for ion and molecular recognition [13]. A remarkable feature of (*S*)-**CCI** (Figure 2a) is that altering the length of the ether spacers in the system allows for the pseudo-*C_3_* symmetry of the oxygen atoms in the cavity and makes the cavity asymmetric. Consequently, the host exhibits colorimetric chiral recognition of simple guest molecules. Figure 2b shows the visual discrimination between enantiomers of amino acids such as phenylglycine (Phg) after a solid (Phg)–liquid ((*S*)-**CCI** in EtOH) two-phase solvent extraction process, where the addition of (*R*)-Phg caused a red-shift of 20 nm with increasing absorption intensity, reflecting the red-to-reddish-violet color change under these conditions, whereas the addition of (*S*)-Phg led to almost no change in the absorption spectra. The chiral discrimination behavior was confirmed *via* HPLC (Figure 2c), where ethanol solutions of (*S*)-**CCI** were stirred with an excess of solid Phg for 12 h at 25 °C. The extracted Phg was analyzed by HPLC on a chiral packing column and pH 1.5 HClO_4_ aq. as an eluent [14]. A straight line obtained in Figure 2c suggests a preferential 1:1 complex. The binding constant of (*R*)-Phg to (*S*)-**CCI** could be determined to be 383 ± 15 M^−l^ by a solubility method [15]. The chiral sensing behavior may be interpreted by the mechanism being as follows: the ethylene oxygens and the indophenolate of (*S*)-**CCI** could stabilize the host–guest complex and restrict the rotation of the guest along C*-NH_3_^+^ axis. Subsequently, the binaphthyl group can act as a minor steric-repulsive group for the (*R*)-isomer and as a major steric-repulsive site for the (*S*)-isomer. However, hosts with two indophenol-type chromophores are sensitive to bulk conditions, indicating that contamination may interfere with colorimetric responses. The proverbial drawback is its labor-intensive synthesis, which leaves little potential for structural variations.

Fluorescence (FL) is a highly effective signal that converts chiral recognition events into discernible outputs through fluorescence enhancement/quenching, excimers, and exciplexes [16]. Versatile signaling modes, such as photoinduced electron transfer, fluorescence resonance energy transfer, and excited-state intramolecular proton transfer, enable numerous designs for the development of fluorescence sensors [17]. For example, fluorescence chiral boronic acid sensors are plausible due to the dynamic covalent bonding nature of the enatioselective interactions between the sensors and chiral analytes such as monosacchraides and tartaric acids [18,19]. However, the optical discrimination of chirality is mainly based on turn-on or turn-off changes in fluorescence [4]. This is because the popular BINOL fluorophore units are insensitive to stimulus-induced color changes, hindering their use as colorimetric chiral sensors that show a noticeable FL color change [13]. In 2021, ensemble-type colorimetric FL sensors were developed by combining green-light-emitting probe (*S*)-**BN** with rhodamine-appended derivative (*R*)-**BNRhd** (Figure 3a) [20]. This structural feature originates from (*R*)-**BNRhd**, where the formyl group serves as the amino acid-binding site, and the rhodamine unit undergoes a ring-opening reaction with Zn^2+^. A 1:1 mixture of (*R*)-**BNRhd** and (*S*)-**BN** in the presence of Zn(OAc)_2_ caused a color change from green to red as the amount of the D-enantiomer of histidine (His) increased (Figure 3b), allowing for quick quantification of the enantiomeric composition.

Unlike conventional fluorophores, compounds with aggregation-induced emission (AIE) display low fluorescence in solution with enhanced emission in the aggregated or solid states [21]. AIE luminogens that respond to a chiral analyte can be regarded as hierarchical chiral sensors. Based on the working mechanism of the conical intersection model [22]. AIE-based chirality sensing systems have been investigated to achieve highly selective and sensitive chiral recognition [23]. However, these systems are mostly based on the differences in the fluorescence intensity between the enantiomers, indicating that tuning the emission color of AIE luminogens through guest chirality is challenging [24]. In addition, AIE-based luminophores with colorimetric response toward guest chirality are rare, possibly because the AIE effect of D–π–A-typed dyes is inversely proportional to the intramolecular charge transfer effect [25]. Nevertheless, Zheng et al. reported an enantiomer excess (*ee*) determination method using the emission wavelength changes to AIE luminogens [26]. The proposed host **TPTA** is a tetraphenylethylene (TPE) derivative incorporating enantiopure amino groups with a bulky cyclohexyl group (Figure 4a), enabling the chiral recognition of phenyl-substituted carboxylic acids as guests because the repulsion strength between the phenyl rings in the host can be tuned *via* the acid–base interactions between the guest analyte and the host. Figure 4b shows the optical sensing of (*S*)-**TPTA** upon the addition of di-*p*-toluoyl-D-tartaric acid (D-**TTA**) in cyclohexane/acetone (98:2 *v*/*v*). Although the *S*-host emitted yellow light with *λ*_em_ of 536 nm under UV light excitation, the addition of D-**TTA** and L-**TTA** induced a change in the emission behavior accompanied by a blue-shift of *λ*_em_ from 536 nm to 457 nm and 483 nm, respectively, resulting in guest-chirality-dependent discernible color emission (Figure 4b). Furthermore, the significant change in the emission maximum wavelength of the host upon varying the *ee* values of the guest carboxylic acid allowed for the determination of enantiomeric purity.

## 4. Molecular Organization

Supramolecular organization is a promising hierarchical approach to bottom-up organization through interactions encoded within the module structures to provide predefined systems. It is driven by not only non-covalent interactions such as hydrogen bonding, van der Waals interactions, coordination bonds, and π–π stacking [27] but also by dynamic covalent bonding interactions, including imine formation, boronate esterification, and disulfide exchange [28]. The indicator displacement assay (IDA) is a well-known effective method for analyte detection, which is based on competitive binding modes of the host/indicator and host/guest ensembles [29]. Anslyn et al. applied IDA for colorimetric sensing using boronic-acid-appended chiral hosts participating in boronate esterification (Figure 5a) [30]. The chelation of metal ions with amino and carboxyl groups contributes to multipoint substrate organization, leading to the desired enantioselectivity (Figure 5b) [31]. This approach was developed for pattern-based recognition using chemometrics (see below).

Another promising self-organization strategy for colorimetric sensing involves the use of doped cholesteric liquid-crystal (LC) films. LCs are partially ordered anisotropic fluids that, from the perspective of thermodynamics, show properties between those of three-dimensionally ordered crystals and those of isotropic liquids, and they have emerged as fascinating photonic materials [32]. When a chiral nonnematic guest molecule induces cholesteric or twisted LC phases in an achiral nematic liquid-crystalline host compound, the aligned films composed of the LC phase reflect light of a particular wavelength, enabling visual detection by tuning the LC within the visible spectrum range. The color is caused by Bragg-like reflections associated with the periodicity of the cholesteric helix and can be observed [33]. Feringa et al. generated colors in LC films aligned on a polyimide-coated glass plate [34]. The structural motif of the LC host material **E7** induced a significant helical twisting power when doped with **D** as the target chiral species (Figure 6), allowing for a direct visual determination of the *ee* of the **D** dopant. Importantly, this insight was applied to an LC-based color test to determine the *ee* of the product produced *via* the catalytic asymmetric 1,4-addition of para-*n*-heptyloxyphenyl-substituted chalcone with diethylzinc. This method allows for the fast and accurate screening of the enantioselectivities of the products in asymmetric catalysts.

The well-tailored organization of organic phosphors in the solid state may result in room-temperature phosphorescence (RTP) based on triplet exciton dissipation [35]. The large Stokes shift in the emission with a long lifetime allows for the detection of the afterglow in the visible region [36]. Zhang et al. focused on the RTP phenomenon and developed afterglow-based solid-state chiral sensing systems [37]. The sensing scheme was based on naphthalimides (**X**) (0.1%, *w*/*w*) easily obtained from the reaction of chiral amino acids with naphthoyl chloride, which were doped into the host (**L**) (Figure 7a). The enantioselective emission phenomena were observed after 254 nm light irradiation at 25 °C, where Ep values were defined as enantiomeric RTP enhancement ratios. Subsequently, 15 amino acids could be reliably discriminated with an Ep value ≥ 3.0 (Figure 7b). This approach, as shown in Figure 7c, showcases the advantages of organic RTP sensing.

## 5. Use of Chemometrics for Self-Organized Colorimetric Sensors

Supramolecular sensors fabricated using synergistic combinations of receptor and optical signaling units undergo multi-validation analysis by structurally varying each unit. To this end, chemometrics has been widely used as a powerful mathematical and statistical method [38,39,40]. Automatic and accurate information collection can allow for the classification and regression of data, enabling a quantitative determination of chirality and the use of prediction techniques, which are often based on machine learning (ML) [41]. As a representative example, a fingerprint-like output was observed through IDAs within a single array using different chiral boronic acid-appended receptors and three indicator dyes, as shown in Figure 8a [42]. Using 96-well plates and a UV/Vis plate reader, the absorbance data for each host-indicator pair ((*S*,*S*)-**BCOMe–BPG**, (*S*,*S*)-**BPh**–**PV**, and (*R*,*R*)-**BPh**-**ML**) were collected and were analyzed by principal component analysis (PCA) [43] to obtain excellent capability for distinguishing the diols and their enantiomers. Furthermore, the datasets of the diol **MPP** with various *ee* values for a given concentration were achieved *via* host–guest combinations of (*S*,*S*)-**BCOMe–BPG**, (*S*,*S*)-**BPh–ML**, and (*S*,*S*)-**BPh–PV** and clustered in smooth curves, and the *ee* values ranged from +1 to −1 (Figure 8b). This analysis enabled the optical response to be resolved statistically. The prediction was carried out *via* a multilayered perception (MLP) network, a type of artificial neural network (ANN) consisting of multiple layers of neurons, with 14 absorbance inputs and 8 processing units in the hidden layers (Figure 8c). ANN is well recognized as a machine learning model inspired by the human brain’s interconnected network of neurons [44]. Minimizing the discrepancy between the input and output data allowed for the accurate determination of the concentration and *ee* of unknown samples of chiral vicinal diols.

α-Cyanostilbenes show unique photophysical properties in which vertical α-cyano substitution on the principal axis in the π-framework regulates intermolecular interactions upon aggregation to tune the emission intensity and wavelength [45]. Kubo et al. studied the effect of the CN group position on the emission properties of D–π–A-typed cyanostilbenes. Ensemble L-***α*NCS-TA** (Figure 9a) showed AIE behavior with an increasing water ratio in EtOH/H_2_O. By contrast, its stereoisomer, L-***β*NCS-TA**, showed typical D–π–A character with aggregation-caused quenching under similar conditions. Such distinctive behavior allowed for the detection of chiral responses to 1,2-cyclohexanediamines (**CHDA**) individually, and the data were collected by varying the fraction of EtOH in the EtOH/THF mixture [46]. These optical responses provide meaningful data processing parameters via linear discriminant analysis (LDA) [47] and artificial neural networks (ANNs). To construct the dataset, the emission data were analyzed by Voigt profile [48] and concatenated into one dataset to estimate the chirality of the analyte. The authors confirmed that the resultant LDA model discriminated between the (1*R*,2*R*) and (1*S*,2*S*) enantiomer pairs using the dimension-reduced axis F1 (Figure 9b). Furthermore, the LDA 2D canonical plot simultaneously classified two types of enantiomer pairs: **CHDA** and 1,2-diphenylethylenediamines (**DPDA**) (Figure 9c) [40]. The LDA model demonstrated precise classifications to learn the intrinsic patterns in the dataset. The ANN algorithm enabled the prediction of ee values for unknown combinations of (1*R*,2*R*)-and (1*S*,2*S*)-**CHDA**. Quantitative analysis and prediction of the ee values of the chiral analytes were successfully conducted using AIE-based chiral recognition, overcoming the low reproducibility of AIE properties.

## 6. Polymer-Based Organizations

The dynamic behavior of two-dimensional polymer scaffolds is attractive for colorimetric chiral sensors, enabling chiral analytes to tune not only their helicity with a low inversion barrier but also their helical pitch through covalent or non-covalent interactions [49]. Maeda et al. proposed helical polymer-based versatile color indicators to assign absolute configurations of chiral amines and determine their ee values [50]. The single-handed helical **M-*h*-poly-1-H** was prepared using the helicity induction and memory strategy (Figure 10a) [51]. The carboxylic groups were modified with (*S*)- and (*R*)-**2a** as putative chiral amine guests using 4-(4,6-dimethoxy-1,3,5-triazine-2-yl)-4-methoxymorpholium chloride as the condensing reagent to give the diastereomeric helices **M-*h*-poly-1-*S*2a** and **M-*h*-poly-1-*R*2a**, respectively (Figure 10b). Dilution of the reaction mixture enabled discrimination of the absolute configuration of **2a** through visual differences in the solution color and fluorescence emission. Similar assays for the naked-eye detection of chirality were performed for various chiral amines, amino alcohols, and amino acid esters upon reaction with **M-*h*-poly-1-H**. The working mechanism can be interpreted as cooperative intramolecular hydrogen bonding between neighboring amide pendants along the helical backbone. These intriguing results allowed for rapid on-site monitoring of the chirality of nonracemic amines to determine their ee values with quantitative accuracy.

In another example of the exploitation of the non-covalent interactions with chiral analytes, Sakai et al. developed poly (phenylacetylene)-bearing chiral amide receptors derived from L-phenylalanine and **PPAPhe** (Figure 11a) [52]. The addition of the L-Leu guest into the THF solution of **PPAPhe** led to an immediate color change from yellow to purple, whereas a yellow-orange solution was observed upon the addition of D-Leu. Such colorimetric discrimination was confirmed *via* the difference of 111 nm between the *λ*_max_ values obtained when both enantiomers were added to the solution. The use of chromophore-containing gels is a plausible hierarchical approach to colorimetric sensing. Iida et al. developed chiral supramolecular organogels composed of riboflavin and melamine (Figure 11b) [53]. The formation of three-point hydrogen binding between **F1** and **M** and the π–π stacking of the flavine chromophore regions led to a yellow helical **F1**/**M** gel with a stoichiometric 2:1 ratio. Enantioselective coloration was observed when a mixture of the **F1**/**M** gel and optically active diethyl tartrates was irradiated for 2 h.

Polydiacetylene (PDA) is a well-known functional polymer. An ordered arrangement in various self-assembled systems undergoes topochemical photopolymerization to form a hierarchically colored phase, which can be applied to visual sensors to detect various species [54]. Liu et al. synthesized L-glutamic acid-terminated 10,12-tricosadiynoic acid (**TCDA-Glu**), which was fabricated into vesicles in water via ultrasonication (Figure 12) [55]. By contrast, supramolecular gels of **TCDA-Glu** can form helical structures in mixed methanol/water solvents. Photopolymerization of these self-assembled structures led to the formation of dispersed nanostructures with a blue color. When tert-butylsulfamide (**TBSA**), which is an important pharmaceutical intermediate, was employed for the visual sensing of chirality, the assemblies with the *S*-enantiomer turned red, whereas those with the *R*-enantiomer remained blue. The colorimetric response to chirality was interpreted based on the enantioselective recognition of the **TBSA** occurring on the surfaces of the assemblies. The synthetic versatility of the monomer enabled the modification of the PDA-based chiral sensors according to the chiral analytes [56].

## 7. Colorimetric Nanoprobes for Chiral Sensing

Noble-metal nanoparticles (NPs) have attracted much attention as hierarchical sensing platforms due to their simplicity and convenient readout in the visible region, enabling their use as colorimetric nanosensors for applications such as environmental monitoring [57], food safety [58], and biosensing [59]. The localized surface plasmon resonance is the basis of the colorimetric response, with the maximum extinction peak frequency depending on the properties of the NPs such as size, shape, and composition [60]. Therefore, modification of the nanoparticle surfaces with the desired chiral selectors is a promising strategy for obtaining chiroptical properties [61]. Gold nanoparticles (AuNPs) have been widely used for colorimetric sensors, owing to their distinctive plasmon resonance, [62] with the tuning of the dispersion and aggregation stages by binding events between the nanoparticles and chiral analytes that lead to a discernible color change accompanied by a shift in the absorption spectra [63]. For instance, the enantioselective aggregation of AuNPs capped with *N*-acetyl-L-cysteine was demonstrated for L/D tyrosine (Figure 13) [64]. Silver nanoparticles (AgNPs) capped with citrate were used to determine the *ee* of D-tryptophan by spectrophotometry [65]. Enantioselective coloration was also applied to β-cyclodextrin-modified AgNPs, enabling a chiral assay for aromatic acid enantiomers [66]. The recognition properties were based on the binding strength between *β*-cyclodextrin and the amino acids. Visual high-throughput screening (HTS) of chiral vicinal diols was also proposed, coupled with the reversibility of boron chemistry [67].

Chiral carbon dots (CDs) are zero-dimensional nanomaterials with either a graphitic or amorphous carbon core and chiral surface [68]. Hierarchically emerging surface-to-volume ratio and conductivity can provide multicolor emission that can serve as an amplifier of the chiral recognition event. Zhang et al. have proposed CDs with a fluorescent/colorimetric dual-mode sensory function toward amino acid analytes [69]. A one-pot hydrothermal reaction of *N*-methyl-1,2-benzenediamine dihydrochloride as the carbon precursor with L-tryptophan (L-Trp) as the chiral ligand yielded the target carbon dots (L-**TCD**s). Spherical structures with an average size of approximately 6.4 nm were observed (Figure 14a), and FT-IR measurements provided characteristic absorption data arising from the -OH and -NH vibrations due to the carboxylic acid and amine groups. Figure 14b shows the colorimetric chiral discrimination behavior of L-**TCD**s toward D/L-Gln in the presence of H_2_O_2_. The probe exhibited ratiometric fluorescence chiral recognition of Glu enantiomers. The role of H_2_O_2_ in chiral sensing was comprehensively investigated, and a working mechanism was proposed: L-**TCD**s can be etched with H_2_O_2_, followed by the participation of amino acid enantiomers that influenced the final particle size of L-**TCD**s. Considering the wide applicability of CD, these results are useful for facile and high-throughput chiral sensing. The synergistic combination of nanomaterials allows for ratiometric sensing for noninvasive quantitative measurements [70]. As a part of this study, nanoprobes composed of blue-emitting carbon dots (**BCD**s) and mercaptopropionic acid-capped CdTe quantum dots (**MRA-QD**s) with inherent chirality were developed. Visual tracking of the enantiomer excess using ratiometric nanoprobes paves the way for the development of low-cost on-site chiral discrimination methods [71]. On the other hand, supramolecular ensembles prepared *via* oleic acid-stabilized quantum dots (**OA-QD**s) and a polyethylene glycol chain with a mandeloyloxy unit were applied to chiral sensing for 2-amino-1,2-diphenylethanol. However, the chiral sensing was only based on fluorescence quenching [72].

The exploitation of gel formation and collapse is a promising approach to the development of nanostructures that serve as naked-eye-detectable platforms for enantiomer discrimination [73]. However, achieving enantioselective color changes remains challenging. In this context, molecular imprinting techniques are effective tools for achieving specific molecular recognition [74]. Qiu et al. proposed photonic crystal gels as molecularly imprinted sensor units for the colorimetric enantiomeric recognition of chiral pyroglutamic acid [75]. Figure 15 represents the construction of pyroglutamic acid-based molecularly imprinted photonic crystal sensor. The SiO_2_ photonics crystal template with an orderly structure prepared *via* the self-assembly vertical deposition method was fabricated on the surface of a glass slide. Meanwhile, the solution of a mixture of L-pyroglutamic acid as a template, functional monomers, cross-linking agents (**EGDNA**), and initiators (**AIBN**) was filled into the interstitial space the SiO_2_ photonics crystal-modified glass slide, which underwent polymerization under UV light. After the reaction, the sandwich-like-structure composites were immersed in 1% hydrofluoric acid to yield a polymerized gel membrane. Removing SiO_2_ and the template yielded pyroglutamic acid-imprinted photonic crystal hydrogel membranes (**MIPH**s). The selective response to analytes was assessed by detecting L-pyroglutamic acid and other interfering substances with similar chemical structures (D-pyroglutamic acid, L-tryptophan, L-phenylalanine, and L-proline). Note that the relative Bragg diffraction peak shift of MIPHs toward L-pyroglutamic acid was around 5 times larger than those of the other four amino acids. The excellent selectivity and good reusability of the nanoprobe make it suitable for use as a chiral probe under practical conditions.

## 8. Conclusions and Future Directions

For further reading about hierarchical chiral sensing, such as gelation and covalent organic framework, an excellent review [76] and primary literature [77] are available. The present review summarizes the strategies for the color sensing of analyte chirality through supramolecular interactions. Colorimetric sensing is a clear and straightforward method for expressing the chirality of an analyte. Avoiding requirements for the use of expensive instruments and cumbersome sample preparation is advantageous for practical use. By contrast, quantitative monitoring is challenging because the colorimetric response is easily subject to contamination. The simplicity of the sensing method is a trade-off for accuracy in determining the enantiomer excess. To overcome this difficulty, color sensing of chirality requires a sophisticated design of systems that enable synergistic communication between optical signals and chirality. Therefore, hierarchical platforms based on noble-metal nanoparticles, quantum dots, and photonic crystals can open new avenues for the construction of visual chirality sensors. Current progress in afterglow materials characterized by long lifetimes is advantageous for chiral sensing with a high signal-to-noise ratio. Alternatively, the use of devices such as smartphones is effective in minimizing the influence of human perceptual differences. The author believes that digital-image-coupled chemometrics will become more effective for accurate chiral sensing, enabling portable detection. In this regard, self-assemblies composed of host molecules and several color indicators have been applied to construct datasets in order to attain accurate chirality sensing by exploiting progress in machine learning. In this way, the chemometrics-assisted colorimetric sensing of chirality has great promise in identifying enantiomers of analytes conveniently without the use of expensive instruments. Considering the ongoing progress in ubiquitous sensing platforms integrated with smartphones and chemometrics-based analysis techniques in various applications in the chemical, pharmaceutical, food, and agricultural fields, breakthrough performance for chiral sensing is anticipated.

## Figures and Tables

**Figure 1 molecules-30-01748-f001:**
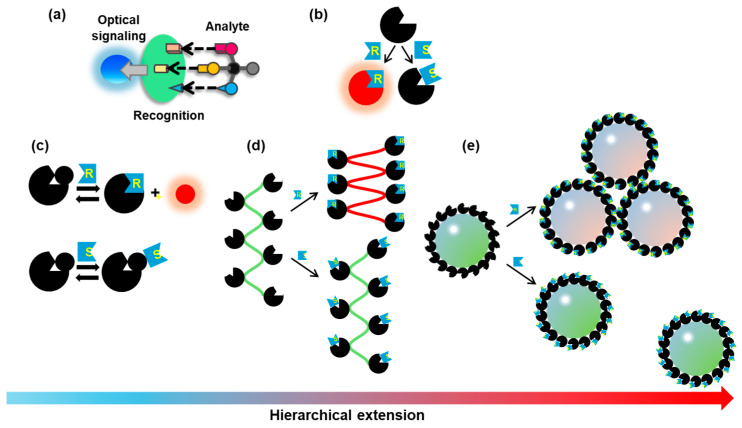
(**a**) Basic design concept for optical sensors for chiral recognition. (**b**) Schematic drawing of colorimetric molecular sensors; a better interaction between the sensor and a chiral analyte leads to a change in color. (**c**) Indicator displacement assay: a host/indicator ensemble undergoes competitive binding behavior with a chiral analyte to release the indicator. (**d**) Host–chiral analyte interactions induce a change in the helical pitch of the π-conjugated polymer backbone. (**e**) Host–chiral analyte interactions on the surface of nanoparticles induce different levels of aggregation, causing a color change. The arrow denotes the direction of structural complexity.

**Figure 2 molecules-30-01748-f002:**
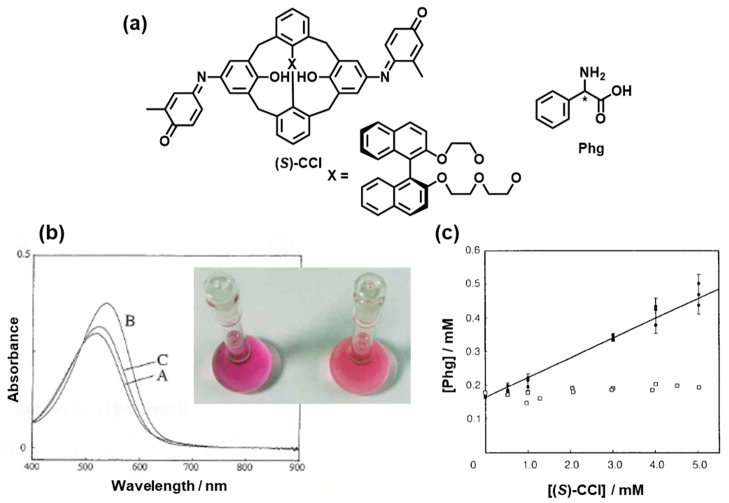
(**a**) Chemical structures of (*S*)-**CCI** and chiral Phg. The * denotes point chirality. (**b**) Visual difference between the Phg enantiomers after the formation of solid (Phg)-liquid ((*S*)-**CCI** in EtOH; 25 °C) two-phase solvent extraction; A: (*S*)-**CCI**, B: (*S*)-**CCI** + (*R*)-Phg), C: (*S*)-**CCI** + (*S*)-Phg. [(*S*)-**CCI**] = 20 μM. In the photograph, (*S*)-**CCI** + (*R*)-Phg and (*S*)-**CCI** + (*S*)-Phg are shown on the left and the right, respectively. (**c**) Extraction of Phg enantiomers, determined by HPLC with EtOH containing various amounts of (*S*)-**CCI** at 25 °C. Reproduced from [14] with permission from Springer Nature.

**Figure 3 molecules-30-01748-f003:**
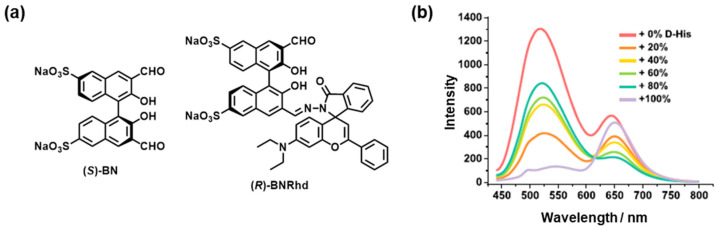
(**a**) Chemical structures of (*S*)-**BN** and (*R*)-**BNRhd**. (**b**) Fluorescence spectra of (*R*)-**BNRhd** (10 μM in DMSO) plus (*S*)-**BN** (10 μM in MeOH) in the presence of Zn^2+^ (40 μM) with 50 equiv. of histidine at various enantiomeric compositions in 50 mM HEPES buffer (pH = 7.4). Reproduced from [20] with permission from the American Chemical Society.

**Figure 4 molecules-30-01748-f004:**
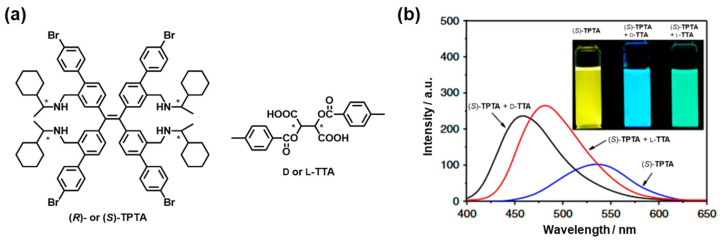
(**a**) Chemical structures of chiral TPE tetramine **TPTA** and chiral **TTA**. The * denotes point chirality. (**b**) Emission spectra of (*S*)-**TPTA** after mixing two enantiomers of **TTA** (**TTA**/(*S*)-**TPTA** = 2, molar ratio) in cyclohexane/acetone (98:2 *v*/*v*). Reproduced from [26]. Licensed by Creative Commons Attribution License (CC BY).

**Figure 5 molecules-30-01748-f005:**
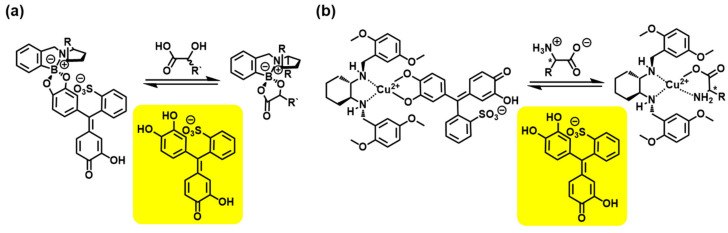
IDA for determining the *ee* of targeted chiral analytes using colorimetric indicators through a boron-based dynamic covalent bond (**a**) or Cu^2+^-coordination (**b**). The yellow-highlighted compound is an indicator of the system. The * denotes point chirality.

**Figure 6 molecules-30-01748-f006:**
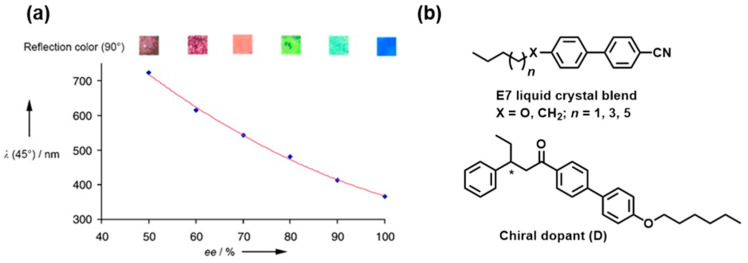
(**a**) Color (90°) and wavelength at an incident angle of 45° of the reflection of **E7** doped with 21 wt.% of dopant D with different *ee* values. The colors depicted are photographs of the LC sample taken perpendicular (90°) to the film surface. (**b**) Chemical structures of the **E7** liquid-crystal blend and chiral dopant (D). The * denotes point chirality. Reproduced from [34] with permission from John Wiley and Sons.

**Figure 7 molecules-30-01748-f007:**
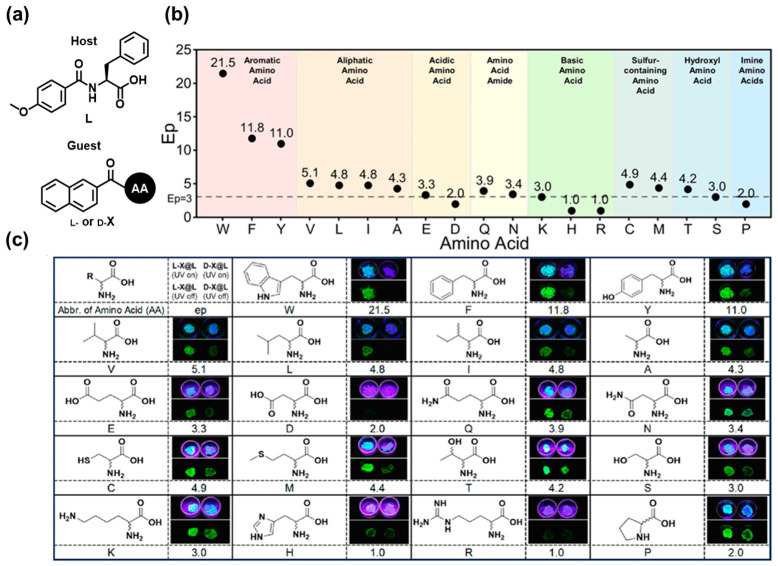
(**a**) Chemical structures of host and guests in chiral recognition of amino acids. (**b**) EP_RTP_ values for chiral amino acids. (**c**) Chemical structures with one-letter abbreviations of the tested amino acids and photographs showing the visual RTP afterglow of doped guest-host samples. Reproduced from [37]. Licensed by Creative Commons Attribution License (CC BY).

**Figure 8 molecules-30-01748-f008:**
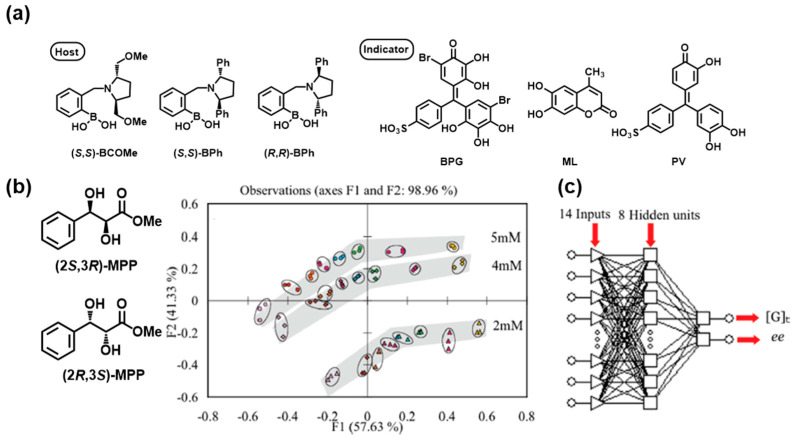
(**a**) Chemical structures of hosts and indicators. (**b**) PCA of diol analytes (2*S*,3*R*)-**MPP** and (2*R*,3*S*)-**MPP** at three different concentrations [2 mM (▲), 4 mM (◆), and 5 mM (●)] for various *ee* values [−1 (yellow), −0.6 (pink), −0.2 (green), 0 (blue), 0.2 (red), 0.4 (orange), 0.6 (brown), and 1 (purple), where 1 is 100% (2*S*,3*R*)-**MPP** and −1 is 100% (2*R*,3*S*)-**MPP**]. (**c**) Artificial multilayer perceptron neural network (MLP) for determining *ee* and analyte total concentration ([G]_t_). Reproduced from [42] with permission from the American Chemical Society.

**Figure 9 molecules-30-01748-f009:**
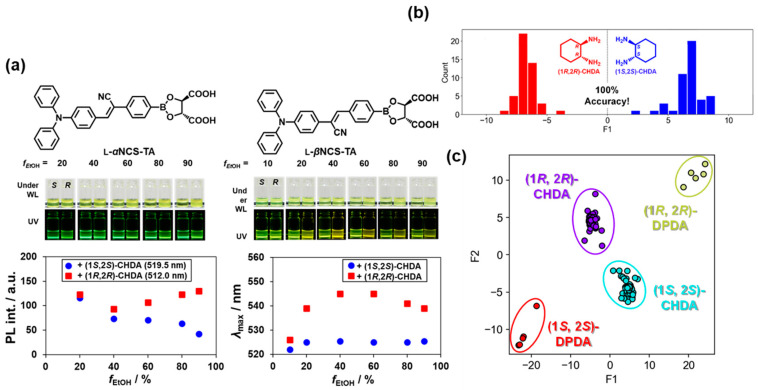
(**a**) Images and plots of emission intensities or emission wavelength of L-***α*NCS-TA** (0.75 mM) or L-***β*NCS-TA** (0.75 mM) with chiral **CHDA** (0.75 mM) upon varying the EtOH fraction in the THF/EtOH mixture. *λ*_em_ values are given in parentheses. (**b**) Training data distributions of (1*S*,2*S*) and (1*R*,2*R*)-**CHDA** ensemble with L**-*α*NCS-TA** and L**-*β*NCS-TA** versus dimension-reduced axis (F1) of LDA. (**c**) LDA canonical plots for analyzing two types of amine enantiomer pairs. Reproduced from [40] with permission from Oxford University Press.

**Figure 10 molecules-30-01748-f010:**
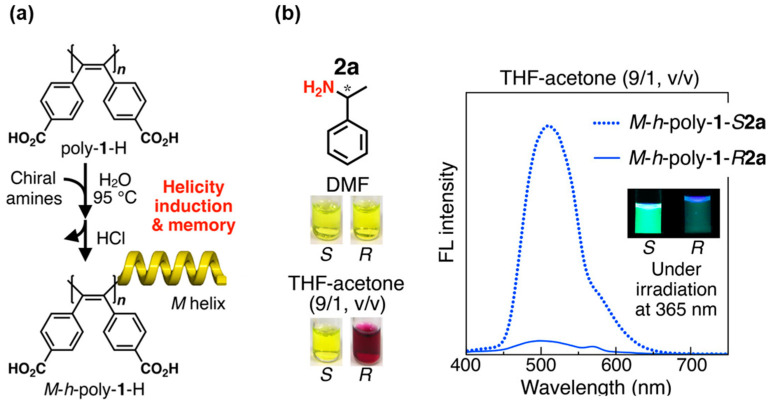
(**a**) Synthesis of left-handed **M-*h*-poly-1-H** with static helical memory, followed by functionalization of the carbonyl pendants with chiral amine **2a**. (**b**) Color of 1.0 mM of **M-*h*-poly-1-*S*2a** and **M-*h*-poly-1-*R*2a** in DMF and THF-acetone (9:1, *v*/*v*) at 25 °C. Fluorescence spectra excited at 350 nm of 0.01 mM of ***M-h-poly-1-S2a*** (dotted) and **M-*h*-poly-1-*R*2a** (solid) in THF-acetone (9:1 *v*/*v*). Inset shows images under irradiation at 365 nm. Reproduced from [51]. Licensed by Creative Commons Attribution License (CC BY).

**Figure 11 molecules-30-01748-f011:**
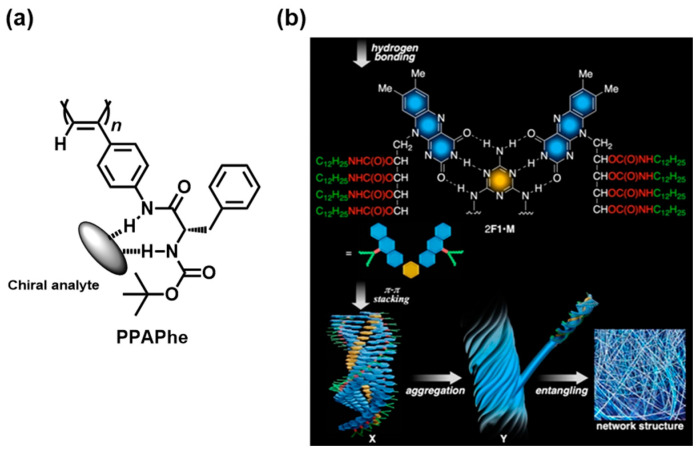
(**a**) Poly (phenylacetylene) with chiral amide receptor pendant. A chiral analyte-induced helical structure may be closely related to chiral sensing. (**b**) Schematic of supramolecular organogels through the hierarchical self-assembly of F1 and M. Reproduced from [53] with permission from John Wiley and Sons.

**Figure 12 molecules-30-01748-f012:**
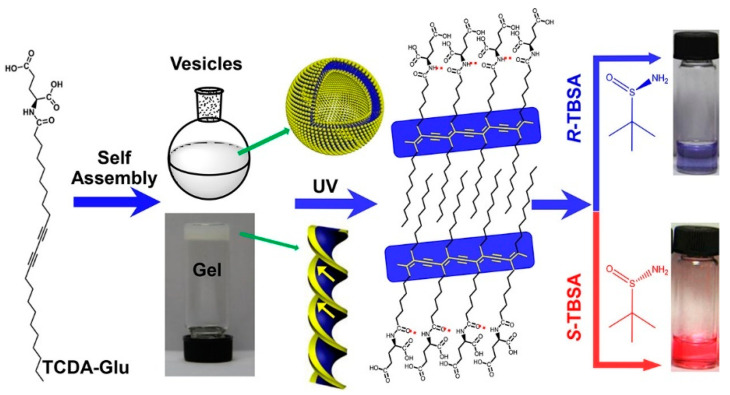
Formation of **TCDA-Glu** vesicles and helices, leading to enantiomeric recognition of **TBSA**. Reproduced from [55] with permission from the American Chemical Society.

**Figure 13 molecules-30-01748-f013:**
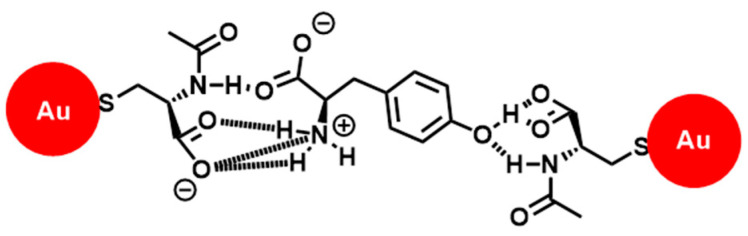
Possible mode of the interaction between L-Tyr and **NALC-Au NPs**.

**Figure 14 molecules-30-01748-f014:**
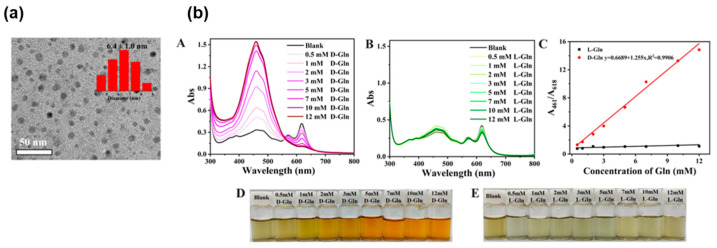
(**a**) TEM image of L-**TCD**s. The inset shows the distribution histogram. (**b**) UV-Vis spectra of L-**TCD**s with different concentrations of L-Gln (**A**) and D-Gln (**B**). (**C**) The relationship between the concentrations of D/L-Gln and *A*_461_/*A*_618_ (*A*_461_ and *A*_618_ are the absorption intensities of **TCD**s at 461 and 618 nm, respectively). Photographs of L-**TCD**s with different concentrations of D-Gln (**D**) and L-Gln (**E**) under sunlight. Reproduced from [69] with permission from the American Chemical Society.

**Figure 15 molecules-30-01748-f015:**
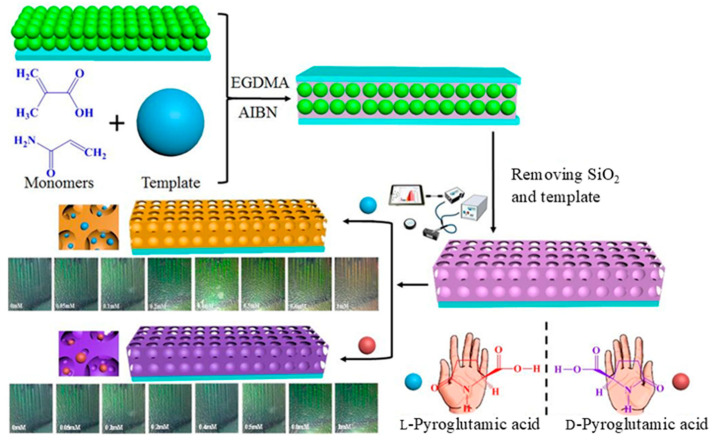
Schematic diagram of preparing pyroglutamic acid-imprinted photonic crystal hydrogel membranes. Reproduced from [75] with permission from Elsevier.

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
