# Peer review of "Colorimetric Visualization of Chirality: From Molecular Sensors to Hierarchical Extension"

_molecules, 2025, doi:10.3390/molecules30081748_

Round 1
Reviewer 1 Report
Comments and Suggestions for Authors
This is a well-written review article on recognition of chirality with molecular probes. I suggest acceptance of the manuscript in its current form. Some minor points may be considered by the author: the quality of some figures are poor; the chiral sensing of tartaric acid, sugar alcohols with organic boronic acids can be briefly mentioned.
Reviewer 2 Report
Comments and Suggestions for Authors
Chiral colorimetric detection is an interesting topic. The article reviews methods such as absorption-based, fluorescence-based, and polymerization-induced emission-based approaches for chiral colorimetric detection, which is beneficial for readers to understand this field. I recommend publishing this article after the following issues are addressed:
- The reference sources cited are rather dated and thus cannot reflect the present landscape of chiral colorimetric detection.
- References summaries in several parts are inadequate. For instance, the citations should have included the paper with DOI: 10.1002/adma.201700296.
- In the design principle segment, the author can incorporate more details about the hierarchical extension. Additionally, the concept map of the hierarchical expansion lacks proper annotations, making it arduous to interpret.
- While elaborating on the work of Kubo et al., the author described the UV - Vis spectra in Fig.2 (b) as "A: (S)-CCI, B: (R)-CCI + (R)-Phg, C: (R)-CCI + (S)-Phg. [(S)-CCI] = 20 µM". Given that, according to the text, (S)-CCI functions as a chiral recognition site for Phg enantiomers, the illustration in Fig.2 (b) conflicts with the corresponding textual account.
- The author should clarify the details of professional terms upon their initial introduction.
- Please provide an elaboration on whether chiral colorimetric detection has the potential to identify enantiomers.
The quality of the English used is good and appropriate.
Reviewer 3 Report
Comments and Suggestions for Authors
I have some general remarks:
- Discussion concern to choosen examples should be more detailed. This is a review, so many people who are not experts in that field will read the paper. In my opinion, Author should explain more carefully the methods of working for those sensors and comment the graphs and figures placed (ie: Figure 2c, Figure 7b and c, Figure 8b and c).
- I would like to see the structures of active spieces made from Hosts/Indicators and Guests.
- There is lack of schemes (ie: Figure 6 - what is the reaction in which the ee was determined? Lines 198-202 and 371-374: there are no schemes or figures for those descriptions)
- reader can not have more information from text bellow the Figure than from the regular text;
- some Figures have unnecessary frames;
- the summary should include a comment on the good and bad sides of the described groups of sensors, not only general statemants;
More detailed remarks:
- Lines 220-221 - which host-indicator pair this part applies to?
- line 238 - should be H2O
- lines 237 and 238 - L-NCS-TA should be in bold
- line 243 and 303 - "via" in italic
- lines 264-273 - compound descriptors should be in bold
- lines 311-312 - what kind of variations do you mean?
Round 2
Reviewer 3 Report
Comments and Suggestions for Authors
The corrections introduced by the Author are sufficient and I have no additional comments.